# Effect of Quaternary Ammonium Salts and 1,2,4-Triazole Derivatives on Hydrogen Absorption by Mild Steel in Hydrochloric Acid Solution

**DOI:** 10.3390/ma15196989

**Published:** 2022-10-08

**Authors:** Yaroslav G. Avdeev, Tatyana A. Nenasheva, Andrey Yu. Luchkin, Andrey I. Marshakov, Yurii I. Kuznetsov

**Affiliations:** A.N. Frumkin Institute of Physical Chemistry and Electrochemistry, Russian Academy of Sciences, 31 Leninskii Prospect, Moscow 119071, Russia

**Keywords:** acid corrosion, corrosion inhibitors, hydrogen penetration into the metal, triazole, steel

## Abstract

The treatment of low-carbon steel items with hydrochloric acid solutions is used in many industrial technologies. This process is accompanied not only by metal corrosion losses, but also by hydrogen absorption by the metal. In this study, the kinetics of hydrogen cathodic reduction on low-carbon steel in 2 M HCl containing corrosion inhibitors, namely, quaternary ammonium salts and a 3-substituted 1,2,4-triazole, have been studied. Adsorption isotherms of corrosion inhibitors on cathodically polarized steel surface have been obtained. XPS data provide valuable information on the composition and structure of protective layers formed on steel in HCl solutions containing inhibitors. The main rate constants of the stages of gaseous hydrogen evolution and incorporation of hydrogen atoms into the metal have been determined. The addition of quaternary ammonium salts or 3-substituted 1,2,4-triazole inhibits the cathodic reduction of hydrogen and its penetration into steel in the HCl solution. 3-substituted 1,2,4-triazole is the most efficient inhibitor of hydrogen absorption. The inhibitory effect of this compound is caused by a decrease in the ratio of the hydrogen concentration in the metal phase to the degree of surface coverage with hydrogen. The maximum decrease in hydrogen concentration in the metal bulk in the presence of the 3-substituted 1,2,4-triazole is 8.2-fold, which determines the preservation of the plastic properties of steel as it corrodes in HCl solutions. The high efficiency of the 3-substituted 1,2,4-triazole as an inhibitor of hydrogen cathodic reduction and absorption results from strong (chemical) adsorption of this compound on the steel surface and the formation of a polymolecular protective layer.

## 1. Introduction

Solutions of hydrochloric acid occupy a special place among acid solutions used for various technological processes. Treatment of carbonate-hydrocarbon-containing formations with hydrochloric acid is used in the oil industry to increase oil recovery; in metallurgical enterprises, thermal scale is removed from steel products with HCl solutions; hydrochloric acid cleaning is also an efficient way to clean the internal and external surfaces of steel products of rust and mineral deposits [1,2,3,4,5,6,7,8,9,10]. Compounds of various nature are widely studied as corrosion inhibitors (CIs) of steels in this medium [11,12,13,14,15,16,17,18,19,20]. Of these, quaternary ammonium salts (QASs) [21,22,23,24,25,26,27,28,29,30] often used as components of industrial mixed CIs are important from a practical point of view. Triazole derivatives are a new promising class of acid corrosion inhibitors for steel that possess unique protective properties [31,32,33,34,35]. These compounds make it possible to protect various steel surfaces in mineral acid solutions at temperatures of up to 200 °C [34,35].

The contact of steel products with corrosive aqueous media is often accompanied not only by corrosion losses of the metal due to its chemical reaction with components of the medium, but also by hydrogen absorption into the metal bulk, which can make the metal brittle, thus significantly impairing its mechanical characteristics [36,37,38,39,40]. Most commonly, the absorption of hydrogen by steels occurs in acid solutions where gaseous hydrogen is a corrosion product released on the metal surface. Molecular hydrogen is formed upon the adsorption of atomic hydrogen that partially penetrates the metal bulk.

The CIs used to protect steels in acid media should not only suppress the general corrosion of the metal, but also hydrogen penetration into the steel. Unfortunately, in many studies of organic corrosion inhibitors for steels, the aspect of their effect on the absorption of hydrogen by a steel is most often ignored, although its mechanical properties are important parameters that ensure the reliable operation of industrial products. In rare studies [41,42,43], it has been noted that nitrogen-containing organic CIs decrease the penetration of hydrogen into steel in acid media.

In view of this, it seems important to identify the regularities of the effect of nitrogen-containing organic CIs on the kinetics of cathodic hydrogen evolution and its penetration into steel. It should be shown how the change in the kinetic parameters of the cathodic reaction caused by inhibitors affects overall corrosion, including estimations of the corrosion rate of steel and the effect on hydrogen absorption, as well as the preservation of mechanical properties by the metal. Detailed discussion of the effect of CI on the kinetics of the cathodic reaction of steel requires features of the mechanism of their action to be identified and reviewed. A mixture of alkylbenzyldimethylammonium chlorides (ABDMA) and an inhibitor (TA) belonging to a promising group of triazole compounds [31,32,33,34,35] were chosen for this study.

## 2. Materials and Methods

### 2.1. Materials

Samples of mild steel (MS, wt.%: 0.05 C, 0.03 Si, 0.38 Mn, 0.09 Ni, 0.04 S, 0.035 P, 0.05 Cr, 0.15 Cu, and 0.16 Al) and spring steel (SS, wt.%: 0.7 C, 1.52 Si, 0.52 Mn, and 0.3 Cr) were used as the working electrodes.

In this study, 2 M HCl aqueous solution was used as the background electrolyte. The solutions were prepared from a concentrated HCl solution of “chemically pure” grade and distilled water. Solutions de-aerated with argon were used in the electrochemical studies. Corrosion tests were carried out in solutions with free access of air. ABDMA, which is a mixture of alkylbenzyldimethylammonium chlorides ([CnH_2_n_+1_N^+^(CH_3_)_2_CH_2_C_6_H_5_]Cl^−^, where n = 10–18), was used as the QAS. Moreover, the TA inhibitor (a 3-substituted 1,2,4-triazole derivative) was used. Due to the low solubility of TA, it was added to the acid solutions as a concentrated solution in ethanol; thus, the concentration of ethanol in the etching solution was 0.24 mol·L^−1^.

### 2.2. Methods

#### 2.2.1. Membrane Test

The rates of hydrogen penetration through the membrane were measured in a Devanathan-Stachurski cell (Figure 1) [44,45]. Mild steel membranes with a diameter of 5 cm and a thickness of 0.1 mm were used. The surface area of the membrane in contact with the electrolyte was 4.25 cm^2^. Both membrane sides were ground with SiC up to #600 grit, then chemically etched in 16% HCl for 1 min and thoroughly washed with distilled water. Subsequently, a palladium film was electrochemically deposited on the membrane’s exit side for 100 s at a constant current density of 25 mA cm^−2^. The palladium plating solution contained 25 g L^−1^ PdCl_2_ and 20 g L^−1^ NH_4_Cl. The solution pH was adjusted to 8.5 by adding the required amount of NH_4_OH. Prior to the tests, the Pd-plated membranes were degassed at room temperature for at least 2 days.

The diffusion part of the cell was filled with 0.1 M NaOH, and the membrane was polarized at a potential of 0.45 V vs. SHE. The hydrogen flux at the exit surface was measured as its Faradic equivalent i_p_ = i − i_bg_, where i_bg_ is the background current density. The background current density was less than 5 × 10^−3^ A m^−2^. A constant potential (E = const) was applied to the working side of the membrane and the external current (i_s_) was measured. The stationary current of hydrogen penetration through the membrane (i_p_) was recorded on the anodically polarized (diffusion) side of the membrane. The potential of the working membrane side was shifted from E = −0.4 V in the positive direction. To fill the hydrogen traps that exist both in the metal bulk and on its surface, before starting the measurements, a potential of −0.4 V was set on the working side of the membrane and maintained until a stationary value of the hydrogen penetration current was established (60 min). The cathodic curves and the i_p_-E plot were obtained at 20 mV increments and 30 min exposure time using an IPC PRO MF potentiostat (Cronas Ltd., Moscow, Russia).

A silver chloride reference electrode and auxiliary platinum electrode were used. All the experiments were carried out at room temperature. Electrode potentials are reported against the SHE.

#### 2.2.2. Electrochemical Impedance Spectroscopy (EIS)

The adsorption of TA was studied by EIS using an IPC-PRO MF potentiostat (Cronas Ltd., Moscow, Russia) combined with a frequency response analyzer (FRA) on a rotating disk electrode (n = 1000 rpm) made of MS steel with a working area of 0.64 cm^2^. Before each experiment, the electrode was cleaned on abrasive wheels with various grain sizes, polished with diamond paste (ACM 0.5/0), and degreased with acetone. To prevent the dissolution of the steel electrode, it was always kept under cathodic polarization (E = −0.30 V). The working and auxiliary platinum electrodes were mounted in a three-electrode cell. A silver chloride reference electrode was used. The electrochemical impedance of a steel electrode was measured in the frequency range of 10 mHz to 3 kHz at an alternating voltage amplitude of 0.020 V. The results of impedance measurements were processed using the Dummy Circuits Solver program, version 1.7. A steel electrode was placed in the acid solution and kept until a stationary impedance spectrum was obtained (no more than 2 h); then, the required concentration of the inhibitor (*C_inh_*) was added to the solution, and the electrode was kept until a stationary spectrum was obtained (no more than 3 h). The experiments were carried out in 2.0 M HCl solutions de-aerated with argon at t = 22 °C.

The surface coverage with the inhibitor (*θ_inh_*) was determined using Equation (1):(1)$$\theta_{inh}=\frac{C_{dl}^{0}-C_{dl}}{C_{dl}^{0}-C_{dl}}$$
where *C_dl_* and *C_dlθ_* are the capacitance of the double electric layer (DEL) of the steel electrode in the background solution, in the inhibited solution, and under conditions of the limiting inhibitor adsorption on the metal, respectively.

The specific capacitance of the DEL was calculated using Equation (2):(2)$$C_{dl}^{sp}=\frac{C_{dl}}{S}$$
where *C_dl_* is the capacitance of the DEL of the steel electrode and *S* is the area of the steel electrode.

#### 2.2.3. Determination of Steel Surface Coverage with an Inhibitor from Stationary Cathodic Current Data

To make the determination of the coverage of MS steel surface with the studied inhibitors more technically feasible, these values were simultaneously calculated from the decrease in the cathodic current (E = −0.30 V) in the presence of the inhibitors (3):*θ_inh_* = (*i*_c,0_ − *i*_inh_)/*i*_c,0_,(3)
where *i_c_*_,0_ is the current density in the background solution and *i_c,inh_* is the current density in the solution with the corresponding additive.

The values of the cathodic currents on MS obtained in the studies described in Section 2.2.1 were used in the calculations. The assumption was made that the additives studied predominantly acted by a blocking mechanism. The results obtained in this way for the adsorption of TA on steel highly correlated with the data obtained by the EIS method (Section 2.2.7). This allowed us to subsequently use the values of *θ_inh_* for ABDMA obtained as the ratio of cathodic currents in the inhibited solution and in the background.

#### 2.2.4. Gravimetric Method

The corrosion rate of SS in 2 M HCl solution was determined from the mass loss of samples (no less than three samples per point) that had a thickness of 0.5 mm and a working surface area of S = 17.6 cm^2^ (4).
*ρ* = Δm·S^−1^ τ^−1^,·100%(4)
where Δm is the change in the sample mass, g; *S* is the sample area, m^2^; and τ is the duration of corrosion tests, h.

Before the start of each experiment, the surface was activated (1 min) with a mixture of acetone and concentrated HCl (10:1) and then wiped dry with a cotton cloth.

The efficiency of inhibitors was estimated as the inhibition factor (5):*Z* = (*ρ*_0_ − *ρ*_inh_) *ρ*_0_^−1^ 100%(5)
where *ρ*_0_ and *ρ*_inh_ are the corrosion rates in the background solution and in the solution with an additive being studied, respectively.

#### 2.2.5. Determination of the Amount of Hydrogen Absorbed by a Metal Using Vacuum Extraction

The hydrogen concentration in the bulk of SS steel was determined using the vacuum extraction method. After the corrosion test, the sample was placed in a vessel that was then evacuated to a residual pressure of 1.33 × 10^−4^ Pa and heated to a temperature of t = 500 °C. The amount of hydrogen released upon heating the sample in the vacuum was estimated from the pressure change in 10 min (P_total_) measured with a McLeod gauge at a constant volume of the evacuated part of the system. The pressure of evolved hydrogen (P_H2_) was calculated from the change in the total pressure (*P_total_*) using Equation (6):*P*_*H*2_ = *P_total_* − *P_correct_*,(6)
where *P_correct_* is the correction for the blank test.

The molar concentration of hydrogen atoms in the bulk of steel (mol cm^−3^) was calculated using Equation (7):(7)$$C_{H}^{v}.=FP_{H2}\;V^{-1}$$
where *K* is a constant related to the volume of the analytical part of the setup, and *V* is the volume of the steel sample, cm^3^.

Data on the volume concentration of hydrogen in the metals are reported with correction for metallurgical hydrogen that amounted to 2.4 × 10^−6^ mol cm^−3^ in the case of the SS.

The degree of steel protection from hydrogen absorption was determined using Equation (8):(8)$${Z^{v}}_{H*}=[(C_{H}^{v}.-\;C_{H}^{v}{}_{,inh}){C_{H^{.}}^{v}}^{-1}]100\%$$
where $C_{H}^{v}.$ and $C_{H}^{v}{}_{,inh}$ are the molar concentrations of hydrogen in the steel bulk after exposure in the background solution and in the inhibited solution, respectively.

#### 2.2.6. Determination of the Mechanical Properties (Ductility) of Steel

The ductility of SS was estimated with an NG-1-3M device based on the number of bends before breakage of band-shaped samples in the initial state (β_0_) and after exposure in 2 M HCl solution (β). The steel ductility was determined using Equation (9):*p* = *β β*_0_^−1^ 100% (9)

The mean value for the SS studied was *β*_0_ = 87.

#### 2.2.7. Voltammetric Studies

Electrochemical measurements were carried out on flat SS samples at t = 25 °C in de-aerated 2 M HCl solutions stirred with a magnetic stirrer. An electrode cleaned and degreased with acetone was kept for 30 min in the test solution; then, anodic and cathodic polarization curves were recorded using an IPC-PRO MF potentiostat (Cronas Ltd., Moscow, Russia) at a dynamic potential scan rate of 0.0005 V s^−1^. The effect of a CI on the rate of the anodic and cathodic processes on the SS in the acid solution was estimated using the inhibition coefficient (10):*γ*_c_= *i_c,_*_0_/*i_c,_*_inh_ and *γ*_a_ = *i_a,_*_0_/*i_a,_*_inh_,(10)
where *i*_c,0_, *i*_a,0_, *i*_c,inh_, and *i*_a,inh_ are the cathodic and anodic current densities in the background solution and in the solution with the additive under study, respectively.

#### 2.2.8. XPS Study of Steel Surface

The quantitative and qualitative compositions of the surface layers formed by TA inhibitor on MS steel were analyzed using X-ray photoelectron spectroscopy (XPS) on an HB100 Auger microscope Omicron ESCA+ instrument (Germany, Taunusstein) equipped with an additional camera for recording XPS spectra. The vacuum in the analytical chamber was no worse than 10^−9^ Torr. An Al anode with 200 W power was used as the excitation source. The pass energy of the analyzer was set to 50 eV. Disk electrodes (MS steel, 10 mm in diameter) served as the samples in the XPS studies. Pretreatment of the samples was performed in the same way as in the electrochemical studies.

The binding energy of electrons (Eb) knocked out from internal shells of atoms was calibrated with respect to the XPS peak of C1s electrons from the vapors of the deposited layers of diffusion oil whose binding energy was assumed to be 285.0 eV. The spectrometer was calibrated with respect to the binding energies Eb of Cu2p3/2 (932.7 eV) and Au4f 7/2 (84.0 eV) from copper and gold samples cleaned with argon ions from surface contaminants. The characteristic peaks of the following elements were measured: C1s, O1s, Fe2p, N1s, and Cl2p. Quantitative estimations were based on the photoionization cross-sections of the corresponding electron shells reported elsewhere [46]. The integral peak intensities were obtained after background subtraction by the Shirley method [47] and by fitting the observed peaks by Gaussian curves with a contribution of the Lorentz component. The integral areas under the C1s, O1s, Fe2p, N1s, and Cl2p peaks were used to calculate the film thicknesses.

An important stage of these studies involves the prolonged (up to 18 min) ultrasonic cleaning of the surface of metal samples from a CI in distilled water or in acid solutions. During this procedure, CI molecules retained on the metal surface by physical forces are removed from the surface of samples pre-exposed to an inhibited acid solution. CI molecules bound to the metal surface by chemical forces are not removed by ultrasonic surface cleaning.

## 3. Results and Discussion

### 3.1. Kinetics of Cathodic Evolution and the Penetration of Hydrogen into Iron in the Presence of Corrosion Inhibitors

Polarization curves were recorded and plots of the rate of hydrogen penetration into steel vs. potential in hydrochloric acid solution and with addition of 0.01–10 mmol organic CIs (ABDMA and TA) were obtained (Figure 2 and Figure 3). As demonstrated, as the CI concentration increased, the cathodic current and the rate of hydrogen penetration into the metal decrease significantly. At low ABDMA concentrations (*C_inh_* = 0.01–0.1 mmol), a sharp decrease in the rates of cathodic hydrogen evolution (*i*_c_) and hydrogen penetration into the metal (*i*_p_) was observed in the entire potential (*E*) region studied, whereas at higher *C_inh_*, the effect of the CI was less pronounced (Figure 2). With an increase in the TA content in the acid solution, the *i*_c_ and *i*_p_ values decreased more smoothly (Figure 3).

### 3.2. Fundamentals of IPZ Analysis of the Dependence of Hydrogen Ion Discharge Rates and Hydrogen Penetration through the Membrane on the Potential [48]

The cathodic evolution of hydrogen on iron in acids occurs by the “discharge—chemical recombination, mixed rate control” or by the “slow discharge—irreversible chemical recombination” mechanisms [48,49,50,51]. The rate constants of the stages of discharge of H^+^ ions and chemical recombination of hydrogen atoms could be determined by IPZ analysis, i.e., by comparison of the cathodic polarization curve and the plot of the current of hydrogen penetration into the metal vs. potential [48]. IPZ analysis could also be applied if a fraction of the electrode surface is blocked by some adsorbed compound, such as a CI [42]. It is assumed in this case that the discharge of H^+^ ions occurs on the metal surface not occupied by adsorbed atomic hydrogen; a corrosion inhibitor added to the solution does not change the mechanism of this reaction. The effect of a corrosion inhibitor on the discharge rate of H^+^ ions (*i*_c_) is described as [52] (11):*i*_c_ = *F k*_c,0_ *a*_H+_ (1 − *θ*_Inh_)^r1^ exp (−*s*_1_*θ*_Inh_) exp(−α*FE*/*RT*),(11)
where *k*_c,0_ is the rate constant H^+^ ion discharge in the background electrolyte, *a*_H+_ is the activity of hydrogen ions, *θ_Inh_* is the coverage of the electrode surface with inhibitor particles, *r*_1_ is the number of adsorption sites occupied by hydrogen ions on the surface, *s*_1_ is a parameter that characterizes the change in the macro properties of the surface layer and takes the possibility of a specific interaction between the activated complex and adsorbate molecules into account, α is the transfer coefficient of the hydrogen ion discharge reaction, *F* is the Faraday constant, *R* is the gas constant, and *T* is the absolute temperature.

Obviously, if we take the “blocking” effect of both atomic hydrogen and inhibitor particles on the rate of discharge of H^+^ ions into account, we obtain (12):*i*_c_ = *Fk*_c_
*a*_H+_ [(1 − *θ*_Inh_)*^r^*^1^ − *θ_H_*] exp(−α*FE*/*RT*),(12)
where *θ_H_* is the surface coverage of the electrode with hydrogen, and *k*_c_ = *k*_c,0_∙exp(−*s*_1_)∙is the rate constant of discharge of hydrogen ions in an inhibitor solution at a constant solution acidity.

If the reaction of chemical recombination of H atoms is irreversible, its rate (*i*_r_) is determined by the expression (12):*i_r_* = *Fk_r_θ*^2^*_H_*(13)
where *k_r_* is the rate constant of chemical recombination of H atoms.

The rate of hydrogen transition through the metal surface (*i_p_*) and its steady-state diffusion in the membrane are described by the relationships (14):*i_p_* = *F*(*k_abc_θ_H_* − *k_des_C^s^_H_*)(14)
and (15):(15)$$i_{p}=\frac{FDC_{H}^{s}}{L}$$
where *k_abs_* and *k_des_* are the constants of absorption and desorption of hydrogen from the metal phase, $C_{H}^{s}$ is the concentration of diffusion-mobile hydrogen in the metal phase near the cathodically polarized membrane surface, *L* is the membrane thickness, and *D* is the diffusion coefficient of hydrogen in the metal.

It follows from Equations (14)–(16):(16)$$\theta_{H}=\frac{k_{des}+\frac{D}{L}}{k_{abs}}C_{H}^{s}=kC_{H}^{s}$$
where *k* is the kinetic-diffusion constant that shows the ratio of the concentrations of hydrogen atoms on the surface and in the metal phase.

Using Equations (12), (13), (15), and (16), we can obtain, for steady-state conditions (*i_c_ = i_p_ + i_r_*) [48,50,53], (17) and (18):(17)$$i_{c}\exp\left(\frac{\alpha FE}{RT}\right)=Fk_{1}a_{H^{+}}{\left(1-\theta_{inh}\right)}^{r1}-\frac{k_{1}a_{H^{+}}kL}{D}\cdot i_{p}$$
(18)$$i_{p}=\frac{D\sqrt{F}}{Lk\sqrt{k_{r}}}\cdot \sqrt{i_{c}-i_{p}}=\frac{D\sqrt{F}}{Lk\sqrt{k_{r}}}\sqrt{i_{r}}$$

By combining Equations (12), (13), (15), and (16), we obtain an expression for calculating the coverage of the steel surface with hydrogen (19):(19)$$\theta_{H}=\frac{-\left(k_{1,i}+\frac{D}{Lk}\right)+\sqrt{{\left(k_{1,i}+\frac{D}{Lk}\right)}^{2}+4k_{r}k_{1,i}{\left(1-\theta_{inh}\right)}^{r1}}}{2k_{r}}$$
where $k_{1,i}=k_{1}a_{H^{+}}\cdot \exp\left(\frac{\alpha FE_{i}}{RT}\right)$ is the formal rate constant of hydrogen ion discharge reactions at the *E_i_* potential. The surface concentration of diffusion-mobile hydrogen in the metal $\left(C_{H}^{s}\right)$ can be calculated from ip values using Equation (15), or from *θ_H_* values using Equation (16).

### 3.3. Calculation of the Rate Constants of the Main Stages of Cathodic Hydrogen Evolution and Penetration into Steel, Coverage of the Steel Surface with Hydrogen, and Surface Concentration of Diffusion-Mobile Hydrogen in the Metal

In all the solutions studied, both in the background solution and in those containing the CIs, IPZ analysis of experimental data can be applied (Figure 2 and Figure 3) because, in accordance with Equation (18), the plots of *i_p_* vs. *i_r_*^0.5^ are almost linear and pass through the origin. As an example, the plots of *i_p_* vs. *i_r_*^0.5^ obtained in the background solution are shown (Figure 4). Comparing the values of $\frac{dE}{dLogi_{c}}$ (and $\frac{dE}{dLogi_{p}}$) and applying the reported method [48], we calculated the transfer coefficients α used to build the plots of $f=i_{c}\exp\left(\frac{\alpha FE}{RT}\right)$ vs. *i_p_*. An example of experimental data processing is shown in Figure 5. Similar plots were obtained for all the solutions and CI concentrations studied.

To calculate the constants *k*_1_, *k_r_*, and *k* according to Equations (17) and (18), the coverage of the metal surface with an inhibitor (*θ_inh_*) must be determined. The EIS method was used to determine *θ_inh_* in solutions containing TA. The impedance spectra of the steel electrode in the background and TA-inhibited 2.0 M HCl solutions presented as Nyquist plots are perfect semicircles described by a simple equivalent circuit that contains the electrical double-layer capacitance (*C_dl_*), reaction resistance (*R_ct_*), and solution resistance (*R_s_*) (Figure 6). An increase in the time of steel electrode exposure in the acid solution containing TA results in an increase in the hodograph radius (Figure 6), which indicates that the inhibitor adsorption occurs slowly over time. The EIS polynomials were calculated using the EIS Methods Applying program [54,55]. In view of this, the electrode was exposed to the inhibited acid solution until a stationary value of its capacitance was reached. The stationary values of *θ*_inh_ calculated according to Equation (1) are given in Table 1, while the dependence of steel surface coverage with the TA inhibitor on its concentration in the corrosive medium (adsorption isotherm) is shown in Figure 7.

The TA adsorption isotherm was obtained in the same solutions by comparing the stationary cathodic currents at E = −0.30 V. The *θ_inh_* values for TA, calculated according to Equations (1) and (3), were nearly the same. For this reason, the *θ_inh_* values for ABDMA were calculated using Equation (3) only. The data are presented in Table 1, and the ABDMA adsorption isotherm is shown in Figure 7.

Using the values of α and *θ_inh_* (Table 1) and assuming *r_1_* = 0.3 [56], from the slope of *f*, *i_p_* straight lines (such as those in Figure 5) and the segment cut off on the y-axis (at *i_p_* = 0), the *k_c,i_* and *k* values were calculated in accordance with Equation (18). Table 1 shows the values of *k* and *k_c,i_* at *E_i_* = −0.3 V.

Using the *k* values obtained and assuming that the stationary diffusion coefficient of hydrogen in the membrane is D = 7.3 × 10^−5^ cm^2^ s^−1^ [57], the values of *k_r_* were determined in accordance with Equation (17) from the slope of the plots of *i_p_* versus *i_r_*^0.5^ (Figure 4) (Table 1).

Thus, the values of kinetic constants of the main stages of hydrogen evolution and penetration into the metal were obtained, both in the background solution and at various *C_inh_* values.

Using the values of the constants (Table 1) and Equation (19), the values of *θ_H_* on cathodically polarized steel surface at E= −0.3 V in 2M HCl background solution and in the presence of CIs were calculated (Table 1). The subsurface hydrogen concentrations in steel $C_{H}^{s}$, calculated according to Equation (15) and Equation (16), satisfactorily agreed and differed no more than by 20% (Table 1 shows the average values of $C_{H}^{s}$).

As follows from Table 1, the addition of corrosion inhibitors to the acid solution significantly decreased the concentration of hydrogen in the metal. The effect of a CI on the surface coverage with hydrogen is ambiguous: the values of θ_H_ can increase with an increase in *C_inh_*. This is due to a decrease in the hydrogen mobilization constant, which, in turn, can be explained by the inhibition of surface diffusion of hydrogen atoms upon the adsorption of inhibitor particles on the metal. At sufficiently high CI concentrations, the values of *θ_H_* decrease because the discharge rate of H^+^ ions decreases to a greater extent than the mobilization rate of H atoms.

The compounds studied not only inhibit corrosion, but also hydrogen absorption, because they decrease the rate constant of discharge of H^+^ ions and increase the *k* constant, i.e., they change the ratio between the *θ_H_* and $C_{H}^{s}$ values (Table 1). The latter effect is explained by the fact that a CI blocks hydrogen absorption centers and hinders the transfer of H atoms from the surface into the metal phase.

TA is a more efficient inhibitor of hydrogen absorption. In fact, the $C_{H}^{s}$ value decreases almost 20-fold at its concentration of 5 mM in the solution (Table 1). A significant decrease in the concentration of diffusion-mobile hydrogen in the metal should benefit the resistance of steel to cracking under mechanical stress.

### 3.4. Effect of Inhibitors on the Corrosion and Mechanical Properties of Steel

The decrease in the hydrogen penetration into the metal by the inhibitors being studied is most noticeable in the case of spring steels that are prone to hydrogen absorption. Indeed, a study of the corrosion of SS steel in 2 M HCl solution showed that these inhibitors reduced the metal mass loss due to corrosion (ρ) and the hydrogen concentration in the metal bulk ($C_{H}^{s}$) (Table 2). It should be emphasized that the plastic properties of the metal (*p*) exhibited almost no change in the presence of TA (Table 2). Therefore, TA is most efficient as an inhibitor of corrosion and hydrogen absorption. It is important that the inhibitory effect of TA is observed not only at 25 °C, but also at 60 °C, which is the optimal temperature for the industrial acid etching of metals. The results obtained agree with the data presented in Section 3.3, i.e., each of the inhibitors decreases both the concentration of diffusion-mobile hydrogen ($C_{H}^{s}$) and the total hydrogen content ($C_{H}^{s}$) in the metal. As a more efficient inhibitor, TA makes it possible to preserve the ductility of SS steel.

### 3.5. Effect of CIs on the Rate of Electrode Reactions on Steel

In the HCl solution, the addition of the CIs being studied are efficient in slowing down the anodic and cathodic stages of the corrosion of SS steel (Figure 8). The anodic polarization slopes (*b_a_*) observed in the presence of ABDMA and TA are 0.15 and 0.16 V, respectively, which are higher than the b_a_ value of 0.12 V observed in the background medium. This effect is more significant for the cathodic reaction because a limiting current is observed in the presence of both CIs, although a value of *b_c_* = 0.16 V is observed in the background medium. The addition of these CIs reduces the rate of anodic ionization of steel, for example, by a factor of 8.8 and 15 in the presence of ABDMA and TA, respectively, at *E* = −0.10 V. The rate of the cathodic reaction at *E* = −0.30 V decreases 9.4- and 13-fold, respectively, in the presence of these CIs. Other things being equal, the effect of the TA inhibitor on the electrode reactions of SS steel is more significant than that of ABDMA. As a result, the smallest mass loss of steel samples is observed in HCl solutions containing TA (Table 2).

### 3.6. Nature of the Adsorption Interaction of the TA Inhibitor with Steel Surface

To understand the reasons for the efficient inhibition of electrode reactions on steel by the TA inhibitor, we have to determine the nature of its adsorption interaction with the steel surface. As shown in Figure 7 (curve 1), the adsorption of TA on the steel surface at medium coverages of the metal surface by the CI obeys the Temkin isotherm (20):*θ**_inh_* = *f*
^−1^ln [*BC_inh_*], (20)
where *θ_inh_* is the surface coverage with the inhibitor, *f* is the surface inhomogeneity factor, *B* is the constant of adsorption equilibrium, and *C_inh_* is the inhibitor concentration in the solution. The calculated values of the parameters are: *f* = 4.25, *B* = 5.31 × 10^5^ L mol^−1^. The free adsorption energy (−ΔG_ads_) was determined using the relationship (21):(21)$$B=\frac{1}{55.5}exp\left[\frac{-\Delta G_{ads}}{RT}\right]$$
and amounted to (−Δ*G_ads_*) = 42 kJ mol^−1^.

The calculated value of the free energy of TA adsorption on steel surface enabled us to assume the chemisorption nature of the interaction between the metal surface and inhibitor molecules because (−Δ*G_ads_*) > 40 kJ mol^−1^ [58]. This mode of interaction between the inhibitor and the surface of steels makes it possible to obtain the highest observed protective effect.

### 3.7. Composition and Structure of the Protective Layers Formed by the TA Inhibitor on Steel Surface

XPS data provide valuable information on the composition and structure of protective layers formed on steel in HCl solutions containing TA. Based on the positions of the complex Fe2p_3/2_ and Fe2p_1/2_ peaks in the XPS spectra of iron and their satellite peaks observed at high binding energies (Figure 9), it may be assumed that a layer consisting of Fe_3_O_4_ (*E_b_* = 710.8 eV) exists on the steel surface. The presence of various types of oxygen is demonstrated by the O1s spectrum that can be decomposed into three peaks from adsorbed water molecules (*E_b_* = 533.5 eV), hydroxyl groups (531.8 eV), and oxygen in the iron oxide lattice (530.3 eV) (Figure 10).

Despite the ultrasonic cleaning of samples in distilled water that removed the physically bound inhibitor layers from the metal surface, the complex XPS spectrum of N1s electrons (Figure 11) indicated that an inhibitor film was present on the steel surface exposed for 24 h in 2 M HCl + 5 mmol TA. The observed N1s spectrum can be decomposed into two peaks (401.4 and 399.5 eV) with a ratio of ~1:3, where the second peak should be attributed to the nitrogen atoms of the triazole group.

Based on the quantitative ratios of surface atoms in the XPS spectra of steel pre-exposed in the inhibited HCl solution with and without subsequent ultrasonic (US) cleaning, it can be concluded that a polymolecular layer of the inhibitor with a thickness above 4 nm was formed on the steel. After the ultrasonic cleaning of samples, only a monolayer of the inhibitor no thicker than 2 nm remained on the steel surface. Such a layer was strongly retained on the metal due to the chemisorptive interaction of the surface iron atoms and nitrogen atoms of the triazole cycle in the inhibitor. The inhibitor layers arranged above the chemisorbed layer were weakly bound to it and to each other by physical interactions and were removed upon such washing. The chemisorbed layer was not removed from the metal surface upon ultrasonic washing and in XPS studies under high-vacuum conditions. The XPS spectrum of the steel surface contained no peak of Cl2p electrons, which indicated the absence of chloride anions in the film. The metal surface under that layer was oxidized to iron oxide when steel samples were washed in the air.

## 4. Conclusions

ABDMA and TA inhibitors hinder the cathodic evolution of hydrogen and its penetration into metal under the cathodic polarization of steel in HCl solution. The adsorption isotherms of both corrosion inhibitors on cathodically polarized steel surface have been obtained. Treatment of experimental data through IPZ analysis [48] made it possible to determine the kinetic constants of the processes, both in the background medium and in the presence of the inhibitors. In the presence of a CI, the reaction rate of the discharge of H^+^ ions decreases, whereas the ratio between the surface coverage with hydrogen and its concentration in the metal phase (kinetic-diffusion constant) increases. As a result, the amount of hydrogen absorbed by steel decreases. TA is the most efficient inhibitor of hydrogen absorption. The addition of 5 mM TA reduces the concentration of diffusion-mobile hydrogen in mild steel almost 20-fold.At the corrosion potential, the TA inhibitor decreases the total hydrogen concentration in spring steel more than 8.2-fold at 25 °C (the degree of steel protection from hydrogen absorption is 87.8%). As a result, the plastic properties of steel exhibit almost no change, and its cracking resistance increases considerably. On raising the solution temperature to 60 °C, the degree of steel protection from hydrogen absorption decreases to 50.8%, but even at this inhibitor efficiency, it increases the resistance of steel to cracking.The TA inhibitor significantly decreases the rate of steel anodic dissolution in the HCl solution. This effect and the hindrance of the rate of cathodic hydrogen evolution determine the efficiency of TA as an inhibitor of the acid corrosion of steels. At a TA concentration of 5 mM, the minimum degree of protection is 94.8%.The high efficiency of TA in slowing down the corrosion of steels and in preserving the ductility of the metal is determined by the mechanism of the inhibitor’s protective action. This compound forms a polymolecular protective layer of triazole molecules up to 4 nm thick on the metal in HCl solutions. The triazole monolayer directly adjacent to the metal is chemically bound to it, whereas the overlying layers are bound to it and to each other due to physical interactions.

## Figures and Tables

**Figure 1 materials-15-06989-f001:**
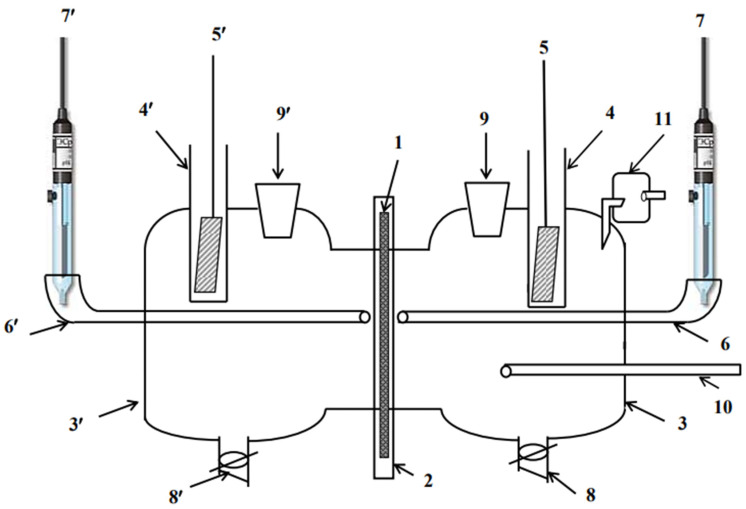
Devanatkhan-Stakhursky electrochemical cell: 1—working electrode (membrane); 2—Teflon holder; 3—working part of the cell; 3′—diffusion part of the cell; 4, 4′—auxiliary electrode cell; 5, 5′—auxiliary electrode; 6, 6’—electrolytic switch; 7, 7′—reference electrode; 8, 8’—tap for draining the solution; 9, 9′—solution input into the cell; 10—argon input into the cell; 11—water seal.

**Figure 2 materials-15-06989-f002:**
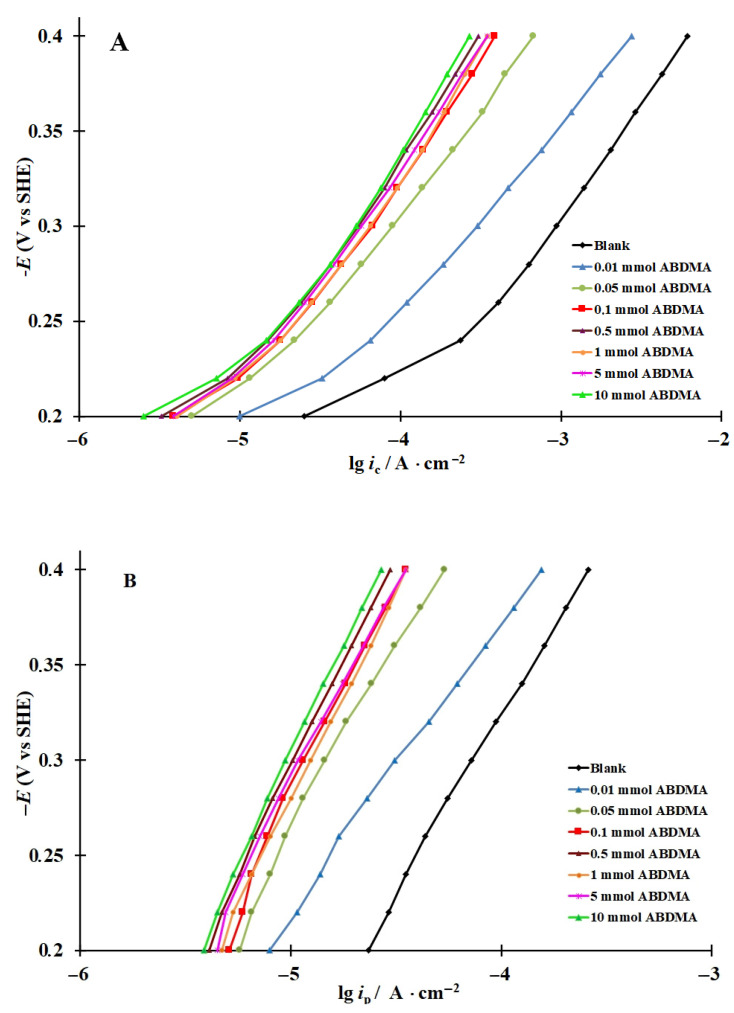
Cathodic polarization curves on steel (**A**) and plots of the rate of hydrogen penetration into steel vs. potential (**B**) in 2 M HCl containing ABDMA.

**Figure 3 materials-15-06989-f003:**
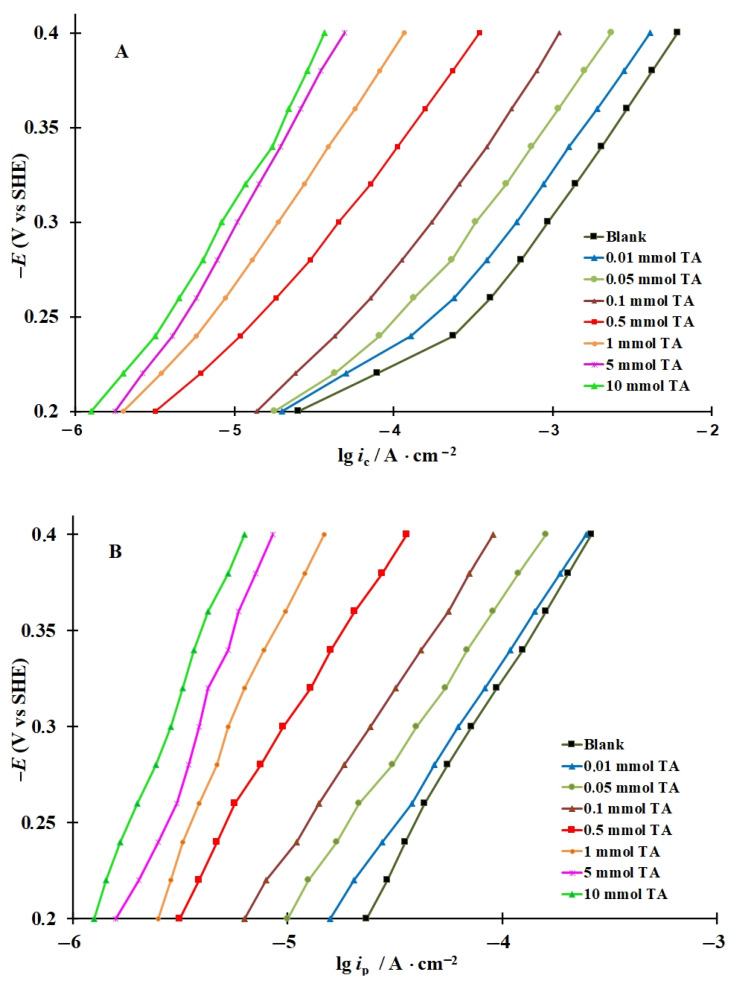
Cathodic polarization curves on steel (**A**) and plots of the rate of hydrogen penetration into steel vs. potential (**B**) in 2 M HCl containing TA.

**Figure 4 materials-15-06989-f004:**
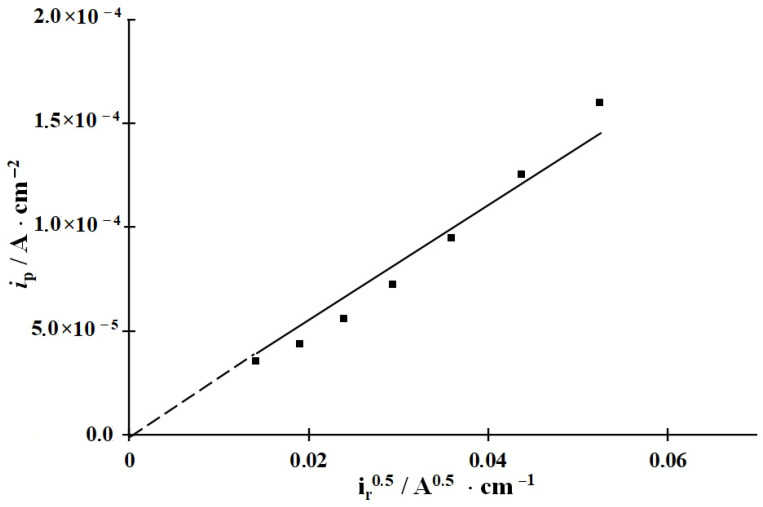
Plot of the current of hydrogen penetration through the membrane vs. the rate of its chemical recombination in 2 M HCl.

**Figure 5 materials-15-06989-f005:**
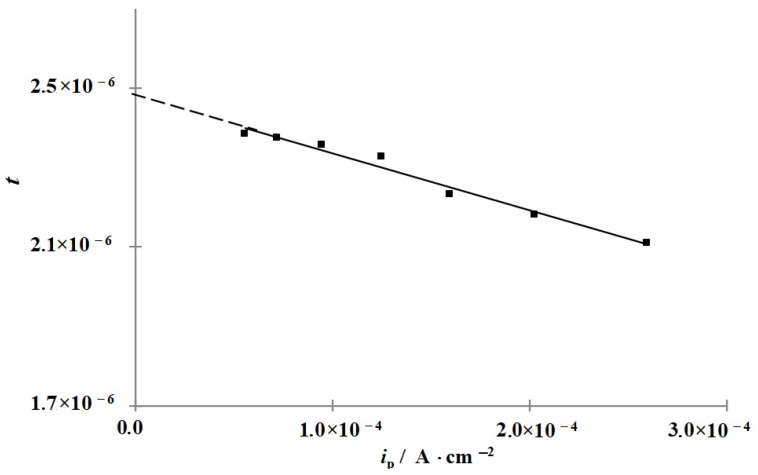
Plot of the function $f=i_{c}\exp\left(\frac{\alpha FE}{RT}\right)$ vs. the current of hydrogen penetration through the membrane in 2 M HCl.

**Figure 6 materials-15-06989-f006:**
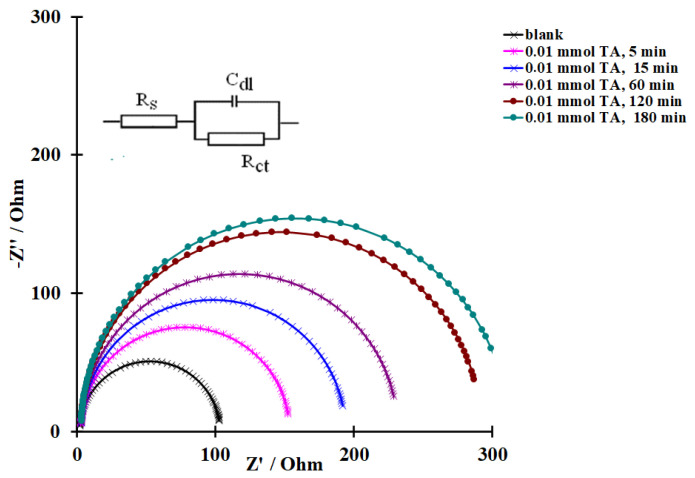
Equivalent electrical circuit and Nyquist plots of a steel electrode (0.64 cm^2^) in 2.0 M HCl recorded after adding 0.01 mmol TA to the solution with various exposure times.

**Figure 7 materials-15-06989-f007:**
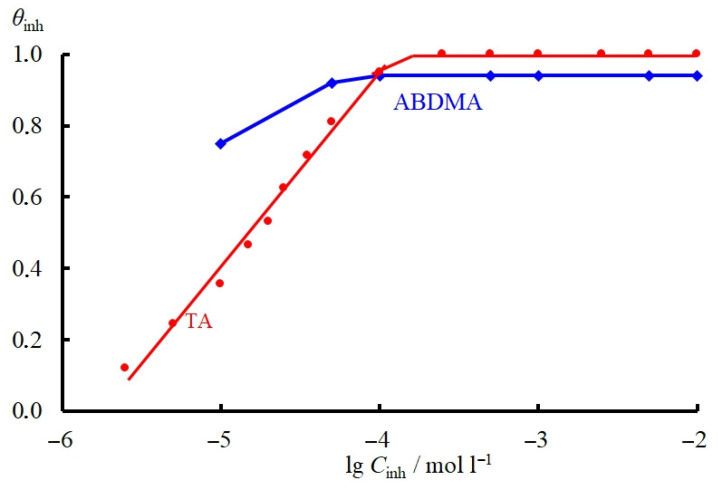
Adsorption isotherm of TA and ABDMA on MS steel (E = 0.30 V) from 2.0 M HCl.

**Figure 8 materials-15-06989-f008:**
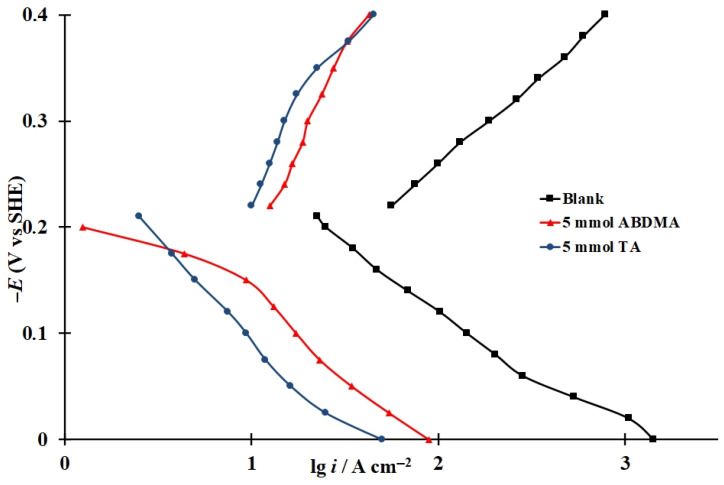
Polarization curves of SS steel at t = 25 °C in 2M HCl in the presence of the following inhibitors ABDMA and TA.

**Figure 9 materials-15-06989-f009:**
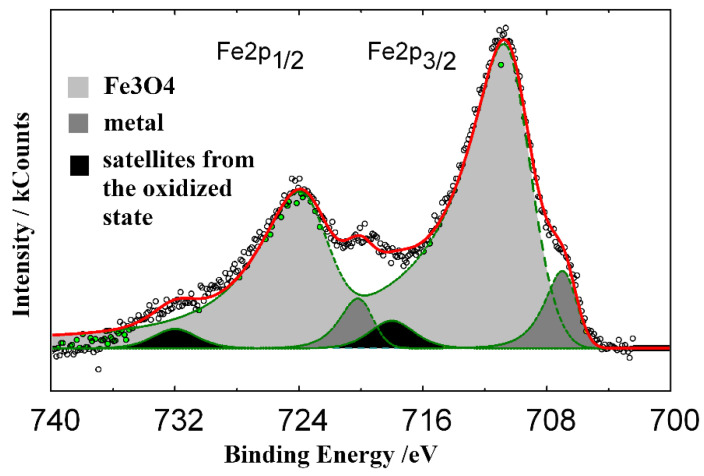
Standard XPS spectrum of Fe2p electrons (spin orbital splitting—doublet) of steel surface after preliminary inhibitor adsorption (2.0 M HCl + 5.0 mmol TA, 25 °C, 24 h).

**Figure 10 materials-15-06989-f010:**
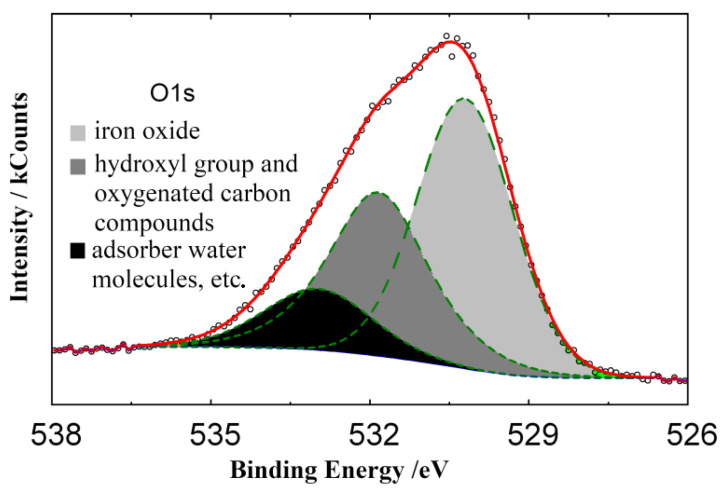
XPS spectra of O1s electrons on the steel surface after preliminary inhibitor adsorption (2.0 M HCl + 5.0 mmol TA, 25 °C, 24 h).

**Figure 11 materials-15-06989-f011:**
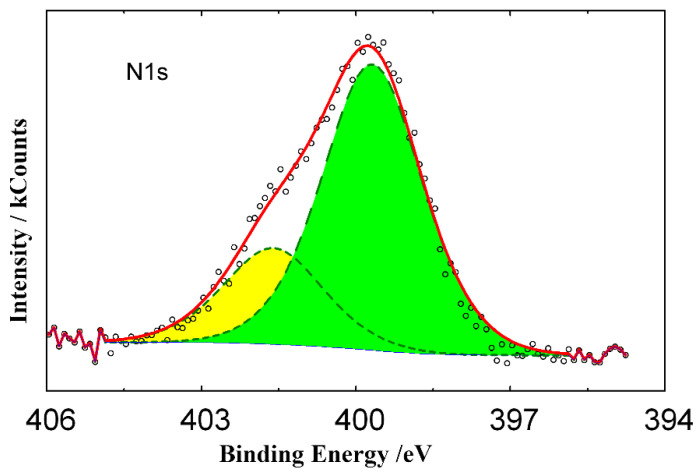
XPS spectra of N1s electrons on the steel surface after preliminary inhibitor adsorption (2.0 M HCl + 5.0 mmol TA, 25 °C, 24 h) followed by washing in an ultrasonic bath.

**Table 1 materials-15-06989-t001:** Kinetic constants, coverage of the metal surface with the inhibitors (*θ_inh_*) and hydrogen atoms (*θ_H_*), and surface concentration of diffusion-mobile hydrogen ($C_{H}^{s}$) under the cathodic polarization (E = −0.3 V) of steel in 2 M HCl solution containing ABDMA and TA.

Inhibitor	*θ* _inh_	*k*_1,i_, mol cm^−2^ s^−1^	*k*, cm^3^ mol^−1^	*k*_r_, mol cm^−2^ s^−1^	*θ*_H_ ×100	$C_{H}^{s}.,\;\times 10^{8}$ mol cm^−3^
Background		9.73 × 10^−9^	3.5 × 10^5^	7.5 × 10^−6^	3.4 ± 0.1	10.0 ± 0.2
ABDMA						
+0.01 mM	0.75	4.7 × 10^−9^	3.8 × 10^5^	6.8 × 10^−6^	2.0 ± 0.1	4.4 ± 0.4
+0.05 mM	0.92	2.0 × 10^−9^	4.9 × 10^5^	5.4 × 10^−6^	1.2 ± 0.1	2.0 ± 0.2
+0.1 mM	0.94	1.5 × 10^−9^	1.3 × 10^6^	1.0 × 10^−6^	2.3 ± 0.1	1.6 ± 0.07
+0.5 mM	0.94	1.3 × 10^−9^	1.8 × 10^6^	5.5 × 10^−7^	2.8 ± 0.1	1.4 ± 0.07
+1 mM	0.94	1.5 × 10^−9^	2.2 × 10^6^	2.9 × 10^−7^	4.2 ± 0.1	1.8 ± 0.06
+5 mM	0.95	1.5 × 10^−9^	2.6 × 10^6^	3.0 × 10^−7^	3.9 ± 0.1	1.5 ± 0.04
+10 mM	0.96	1.4 × 10^−9^	3.0 × 10^6^	2.9 × 10^−7^	3.4 ± 0.1	1.3 ± 0.04
TA						
+0.01 mM	0.42	7.2 × 10^−9^	4.7 × 10^5^	2.5 × 10^−6^	4.6 ± 0.1	8.9 ± 0.5
+0.05 mM	0.72	5.0 × 10^−9^	8.9 × 10^5^	9.1 × 10^−7^	5.6 ± 0.2	5.6 ± 0.3
+0.1 mM	0.86	3.2 × 10^−9^	9.2 × 10^5^	1.1 × 10^−6^	3.7 ± 0.1	3.4 ± 0.2
+0.5 mM	0.96	1.2 × 10^−9^	2.4 × 10^6^	4.0 × 10^−7^	3.0 ± 0.1	1.4 ± 0.07
+1 mM	0.98	6.3 × 10^−10^	2.4 × 10^6^	3.9 × 10^−7^	1.9 ± 0.1	0.75 ± 0.1
+5 mM	0.99	1.1 × 10^−10^	2.5 × 10^6^	6.0 × 10^−7^	1.1 ± 0.1	0.55 ± 0.5
+10 mM	0.99	3.4 × 10^−10^	3.3 × 10^6^	4.4 × 10^−7^	1.1 ± 0.1	0.41 ± 0.3

**Table 2 materials-15-06989-t002:** Effect of 5 mmol of nitrogen-containing inhibitors on the corrosion, hydrogen absorption, and ductility of SS steel in 2 M HCl.

Inhibitor	*t*, °C	*ρ*, g m^−2^·h^−1^	*Z*_cor_, %	$(C_{H}^{v}.),\;\mathrm{mol}\;{\mathrm{cm}}^{-3}$	*Z^v^_H_*_*_, %	*p*, %
Background	25	11.4	-	3.2 × 10^−5^	-	- *
ABDMA	25	6.36	44.2	1.2 × 10^−5^	62.7	-
TA	25	0.59	94.8	3.9 × 10^−6^	87.8	100
Background	60	48.7	-	1.8 × 10^−5^	-	-
ABDMA	60	16.5	66.1	6.9 × 10^−6^	61.3	-
TA	60	2.4	95.1	8.8 × 10^−6^	50.8	91

* Total loss of ductility.

## Data Availability

Not applicable.

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
