# Peer review of "Effect of Quaternary Ammonium Salts and 1,2,4-Triazole Derivatives on Hydrogen Absorption by Mild Steel in Hydrochloric Acid Solution"

_materials, 2022, doi:10.3390/ma15196989_

Round 1
Reviewer 1 Report
Comment 1: Title should be revised and improved.
Comment 2: Qualitative informations are missing in abstract. Abstract should be concise and the authors need to improve with more specific short results.
Comment 3: All acronyms should be introduced at first place of appearance in the text. For example XPS, ... in the abstract section.
Comment 4: Numbers of equation should be mentioned in the text.
Comment 5: In Figures 2, 3, 4, 5, 7 and 8, for numbers, commas must be replaced by dots.
Comment 6: Figure 6 was not fitted well, the author should fit the impedance data using such as Zview soft and so on.
Comment 7: Axis in the Nyquist diagrams (Figure 6) have to isometrics.
Comment 8: Report the standard deviation and errors bars in Table 1. The following reference should added. Inorganic Chemistry Communications. 115 (2020) 107858 (https://doi.org/10.1016/j.inoche.2020.107858)
Comment 9: Conclusion is too, conclusion should be improved.
Comment 10: Level of English is good however in a few places some syntax errors are present. At some places two or more words joined together that should be corrected.
Comment 11: Compare your results with literature ones.
Comment 12: The introduction section should be modified and improved though citing recent references (2021 and 2022) related studies and indicating the novelty of the study compared to the carried works. The following references should added.
ü Journal of Colloid and Interface science. 574 (2020) 43-60 (https://doi.org/10.1016/j.jcis.2020.04.022).
Author Response
The authors are grateful to the Reviewer for valuable comments
Please see the attachment

Reviewer 2 Report
The title of the paper is too long
(Organic corrosion inhibitors) is a very broad title, it seems like a review article title. The title should be specific and informative.
Abstract
The authors should re-write the abstract. The HCl should be mentioned. The values of inhibition efficiency should be mentioned. Information about the adsorption behavior should be added.
Keywords are redundant and unsuitable.
Introduction
The literature review is very poor. Authors should add recent studies that cover using of similar compounds as corrosion inhibitors for mild steel.
All superscripts & subscripts should be adjusted e.g. line 73.
SEM analyses are required to show the morphology of steel surface before and after the inhibition process.
The inhibition process should be studied at different values of temperature in order to determine and discuss to thermodynamic parameters.
It is preferable to use PDP technique to study the corrosion kinetics.
It is highly recommended to perform theoretical studies i.e. DFT calculations and MD simulations.
Author Response

(The authors gave the same response as above.)

Reviewer 3 Report
This paper studied the kinetics of cathodic reduction of hydrogen on low-carbon steel in a hydrochloric acid solution containing inhibitors. This paper is well written. The data is reliable and compressive. Therefore, it is suggested to be published in “Materials”. There some revisions should be addressed, as follows:
1. Why do you choose 2 M HCL as the background electrolyte?
2. The detail size of the membranes should be given.
3. The samples after corrosion can be given.
4. High resolution images are better to replace Figure 1,2,3.
5. More references can be cited from “Materials”.
Author Response

(The authors gave the same response as above.)

Round 2
Reviewer 2 Report
The authors should revise the English language of the manuscript. There are many grammatical and punctuation errors, such as:
Line 16: “namely quaternary ammonium salts and a 3-substituted 1,2,4-triazole, has been” please correct has to have
Line 17: “Adsorption isotherms of corrosion inhibitors on cathodically polarized steel surface have” please correct this phrase (surface to surfaces) OR (have to has)
Line 19: “Addition of quaternary ammonium” should be (The addition)
Line 58: “protect various steels” please correct e.g. (steel surfaces or sheets of steel)
Line 65: “penetrates into the metal bulk”. Please remove (into).
I am still not happy with the title. It is still not informative. It should contain the main terms, such as corrosion and inhibitors.
The authors didn't respond positively to comments 6 to 9. Since the authors are working in the field of corrosion inhibitors, and since they claim that their inhibitors have good efficiency against the corrosion phenomena, they must use all the powerful tools to prove their claim. SEM & PDP is an essential tools in the field of corrosion inhibitors.
Author Response

(The authors gave the same response as above.)
